# A Spectral Framework for Assessing the Geodesic Distance Between Graphs

## Abstract

This paper presents a spectral framework for quantifying the differentiation between graph data samples by introducing a novel metric named Graph Geodesic Distance (GGD). For two different graphs with the same number of nodes, our framework leverages a spectral graph matching procedure to find node correspondence so that the geodesic distance between them can be subsequently computed by solving a generalized eigenvalue problem associated with their Laplacian matrices. For graphs of different sizes, a resistance-based spectral graph coarsening scheme is introduced to reduce the size of the larger graph while preserving the original spectral properties. We show that the proposed GGD metric can effectively quantify dissimilarities between two graphs by encapsulating their differences in key structural (spectral) properties, such as effective resistances between nodes, cuts, the mixing time of random walks, etc. Through extensive experiments comparing with the state-of-the-art metrics, such as the latest Tree-Mover's Distance (TMD) metric, the proposed GGD metric shows significantly improved performance for graph classification and stability evaluation of GNNs, especially when only partial node features are available.

## 1 Introduction

In the era of big data, comparison and distinction between data points are important tasks. A graph is a specific type of data structure that represents the connections between a group of nodes or agents. Comparing two graphs often involves using a pairwise distance measure, where a small distance indicates a high structural similarity and vice versa. To understand the generalization between distribution shifts, it is important to use an appropriate measure of divergence between data distributions, both theoretically and experimentally (Chuang et al., 2020). Determining suitable distance metrics for non-Euclidean data entities, like graphs with or without node attributes, which are fundamental to many graph learning methods such as graph neural networks (GNNs), remains a significant challenge, even though distance metrics for data points in Euclidean space are readily available. The need to develop new analytical techniques that allow the visualization, comparison, and understanding of different graphs has led to a rich field of research study (Haslbeck & Waldorp, 2018). This study dives into the exploration of a novel framework for computing geometric distances between graphs, which can be immediately leveraged for many graph-based machine learning (ML) tasks, such as graph classification or the stability evaluation of GNNs.

Many distance metrics for comparing graphs have previously been proposed (Borgwardt et al., 2020). Some of them are merely based on graph local structures (Tam & Dunson, 2022; Haussler et al., 1999; Xu et al., 2013; Zhu et al., 2020; Fernández & Valiente, 2001; Bunke & Shearer, 1998), whereas others exploit both graph structural properties and node attributes (Shervashidze et al., 2011; Morris et al., 2019). For example, the Graph Edit Distance (GED) has been proposed to measure the distance between graphs considering the number of changes needed to match one graph to another (Sanfeliu & Fu, 1983; Gao et al., 2010; Li et al., 2017); Distance metrics based on the graph kernel have also been investigated (Shervashidze et al., 2011; Vishwanathan et al., 2010), such as the Wasserstein Weisfeiler-Leman metric (WWL) (Morris et al., 2019) and the Gromov–Wasserstein metric (Mémoli, 2011), which allow computing graph distances based on low-dimensional graph representations or optimal transport (OT) (Titouan et al., 2019; Chapel et al., 2020), leading to the development of the state-of-the-art graph distance metric called TMD (Chuang & Jegelka, 2022).

However, the existing graph distance metrics have distinct limitations. For example, the GED metric can capture local node or edge changes but struggles with global perturbations (Sanfeliu & Fu, 1983; Gao et al., 2010; Li et al., 2017); the WWL and TMD metrics heavily rely on node features (attributes) for calculating the distance between graphs, leading to degraded performance when only partial node features are available (Rossi et al., 2022; Chen et al., 2022).

To address these limitations of prior methods, we propose the Graph Geodesic Distance (GGD) metric, a novel approach that leverages spectral graph theory and Riemannian geometry to effectively quatify topological distance between graphs. This framework handles graphs of the same size by using spectral graph matching to determine node correspondence and computes distances on a Riemannian manifold of modified Laplacian matrices. We show that the proposed GGD metric can theoretically capture key structural (spectral) dissimilarities between two graphs, such as mismatches in Laplacian eigenvalues/eigenvectors, cuts, effective-resistance distances, etc.

One distinct advantage of the proposed GGD metric is its capability to compute distances between graphs based on their spectral (structural) properties, while including node feature information into our framework can further improve the accuracy. Therefore, GGD is suitable for analyzing many real-world graphs that may only have partial or even no node features. Moreover, the proposed framework for computing GGDs is more computationally efficient than existing OT-based metrics, such as the TMD metric.

Our empirical results show that GGD can effectively measure the dissimilarities between graphs: (1) support vector classifiers (SVC) using GGDs perform competitively with state-of-the-art GNN models and graph kernels on graph classification benchmarks; (2) we demonstrate that the GGD metric allows us to quantify the stability of GNN models for graph classification tasks by checking whether two graphs with a small GGD will lead to a significant dissimilarity in the GNN output embeddings. We also show that the GGD metric has a better correlation with established GNN outputs compared to the state-of-the-art TMD metric (Chuang & Jegelka, 2022) when only partial node features are available: up to a $10\%$ accuracy gain and a $9\times$ runtime speedup have been achieved in various graph classification tasks.

## 2 EXISTING GRAPH DISTANCE METRICS

**Graph Edit Distance (GED)** For non-attributed graph data, a common and simple distance metric is GED. (Sanfeliu & Fu, 1983; Gao et al., 2010). Given a set of graph edit operations, also known as elementary graph operations, the GED between two graphs $G_1$ and $G_2$, written as $\text{GED}(G_1, G_2)$, can be defined as:

$$\text{GED}(G_1, G_2) = \min_{(e_1,\dots,e_k)\in\mathcal{P}(G_1,G_2)} \sum_{i=1}^{k} c(e_i),\tag{1}$$

where $\mathcal{P}(G_1, G_2)$ denotes the set of edit operations transforming $G_1$ into a graph isomorphism of $G_2$, $c(e_i)$ is the cost of edit operation $e_i$. The set of elementary graph edit operators typically includes node insertion, node deletion, node substitution, edge insertion, edge deletion, and edge substitution.

**Tree Mover's Distance (TMD)** TMD is a pseudometric for measuring distances between simple graphs, extending the concept of WWL to multisets of tree structures (Chuang & Jegelka, 2022). By progressively adding neighboring nodes to the previous node at each level, we obtain the computation tree of a node. These tree structures are crucial in graph analysis (Weisfeiler & Leman, 1968; Pearson, 1905) and graph kernels (Ramon & Gärtner, 2003; Shervashidze et al., 2011). TMD uses hierarchical optimal transport (HOT) to analyze these computational trees from input graphs. For a graph $G = (V, E)$ with node features $f_v \in \mathbb{R}^s$ for node $v \in V$, let $T_v^1 = v$, and $T_v^L$ be the depth-L computation tree of node $v$. The multiset of these trees for $G$ is $T_G^L = \{T_v^L\}_{v \in V}$. The number and shape of trees must match to calculate optimal transport between two multisets of trees. If multisets are uneven, they are augmented with blank nodes. For multisets $T_p$ and $T_q$, the augmenting function $\sigma$ adds blank trees to equalize their sizes. A blank tree $T_\mathbb{0}$ has a single node with a zero vector feature $\mathbb{0}_p \in \mathbb{R}^s$:

$$\sigma : (T_p, T_q) \to \left( T_p \cup T_\mathbb{0}^{\max(|T_q|-|T_p|,0)}, T_q \cup T_\mathbb{0}^{\max(|T_p|-|T_q|,0)} \right).\tag{2}$$

Let $X = \{x_i\}_{i=1}^k$ and $Y = \{y_i\}_{j=1}^k$ be two data multisets and $C \in \mathbb{R}^{k \times k}$ be the transportation cost for each data pair: $C_{ij} = d(x_i, y_j)$, where $d$ is the distance between $x_i$ and $y_j$. The unnormalized Optimal Transport between $X$ and $Y$ can be defined as:

$$\mathrm{OT}_d(X, Y) := \min_{\gamma \in \Gamma(X,Y)} \langle C, \gamma \rangle, \quad \Gamma(X, Y) = \left\{ \gamma \in \mathbb{R}_+^{m \times m} \mid \gamma \mathbb{1}_m = \gamma^\top \mathbb{1}_m = \mathbb{1}_m \right\}. \quad (3)$$

Here $\Gamma$ is the set of transportation plans that satisfies the flow constrain $\gamma \mathbb{1}_m = \gamma^\top \mathbb{1}_m = \mathbb{1}_m$. (Chuang & Jegelka, 2022).

The distance between two trees $T_p$ and $T_q$ with roots $r_p$ and $r_q$ is defined recursively:

$$\mathrm{TD}_w(T_p, T_q) := \begin{cases} \left\| f_{r_p} - f_{r_q} \right\| + w(L) \cdot \mathrm{OT}_{\mathrm{TD}_w}\left( \sigma\left(T_{r_p}, T_{r_q}\right)\right), & \text{if } L > 1 \\ \left\| f_{r_p} - f_{r_q} \right\|, & \text{otherwise} \end{cases} \quad (4)$$

where $L$ is the maximum depth of $T_p$ and $T_q$, and $w$ is a depth-dependent weighting function. Subsequently, the concept of distance from individual trees is enlarged to entire graphs. For graphs $G_1$ and $G_2$, with multisets $\mathbf{T}_{G_1}^L$ and $\mathbf{T}_{G_2}^L$ of depth-L computation trees, the Tree Mover's Distance is:

$$\mathrm{TMD}_w^L(G_1, G_2) = \mathrm{OT}_{\mathrm{TD}_w}(\sigma(\mathbf{T}_{G_1}^L, \mathbf{T}_{G_2}^L)). \quad (5)$$

## 3 GGD: A Geodesic Distance Metric for Graphs

**Modified Laplacian matrices on the Riemannian manifold** One way to represent a simple connected graph is through its Laplacian matrix, which is a Symmetric Positive Semidefinite (SPSD) matrix. Graph representation using adjacency and Laplacian matrices is briefly discussed in Appendix A.2. Adding a small positive value to each diagonal element allows us to transform the original Laplacian matrix into a Symmetric Positive Definite (SPD) matrix, which is referred to as the **Modified Laplacian Matrix** in this work. In Appendix A.8, we describe the effect of this small value on the GGD calculation. We can then consider the cone of such modified Laplacian matrices as a natural Riemannian manifold (Lim et al., 2019), where each modified Laplacian, having the same dimensions (same number of rows/columns), can be regarded as a data point on this Riemannian manifold (Vemulapalli & Jacobs, 2015; Pennec et al., 2006). Details about the Riemannian manifold are provided in Appendix A.3. Finally, the geodesic distance between two graphs can be defined as the shortest path distance on the Riemannian manifold, assuming their node correspondence is known in advance. This approach is more appropriate than directly comparing the graphs in Euclidean space (Lim et al., 2019; Crane et al., 2020; Huang et al., 2015). We will later demonstrate (Section 4.3) that such a geodesic distance metric can effectively capture structural (spectral) mismatches between graphs.

**A two-phase spectral framework for computing GGDs** Before computing GGDs, it is necessary to establish the node-to-node correspondence between two graphs. This can be achieved by leveraging existing graph-matching techniques (Livi & Rizzi, 2013; Emmert-Streib et al., 2016; Caetano et al., 2009). The proposed GGD metric can be computed in the following two phases. **Phase 1** consists of a spectral graph matching step, using combinatorial optimization with the eigenvalues/eigenvectors of the graph adjacency matrices to identify the approximate node-to-node correspondence. **Phase 2** computes the GGD between the modified Laplacian matrices of the matched graphs by exploiting generalized eigenvalues. An algorithmic flow is provided in Appendix A.1 to ensure a clear understanding of the process. The proposed GGD metric differs from previous OT-based graph distance metrics in its ability to accurately represent structural discrepancies between graphs, enabling us to uncover the topological variations between them more effectively. Since only the graph Laplacian (adjacency) matrix is required to calculate the GGD, our metric can even work effectively for graphs without node feature information.

**A motivating example** Let's consider a simple graph $G_1$, characterized by an almost ring-like topology, as shown in Figure 1. We also create two other graphs $G_2$ and $G_3$ by inserting an extra edge into $G_1$ in different ways. Note that the additional edge in $G_3$ will have a greater impact on $G_1$'s global structure since it connects two further nodes.

We compute the normalized distances (the largest distance always equals one) between the aforementioned three graphs using different metrics (GED, TMD, and GGD) and report the results in Table 1.

Table 1: Distance between graphs with simple perturbations.

| Graph pairs | Distance metrics (Normalized) | | | |
| --- | --- | --- | --- | --- |
| | GGD | TMD with NF, L = 4 | TMD without NF, L = 4 | GED |
| $G_1, G_2$ | 0.623 | 0.689 | 0.970 | 1 |
| $G_1, G_3$ | 0.855 | 0.711 | 1 | 1 |
| $G_2, G_3$ | 1 | 1 | 0.333 | 1 |

As observed, $G_2$ and $G_3$ have distances similar to $G_1$ when the TMD metric is adopted without using node features (NFs). On the other hand, the TMD metric can produce similar results as the proposed GGD metric when node features are fully utilized. Not surprisingly, the GED always produces the same distances since only one edge has been added. The above results imply that the GED and TMD (without using NFs) metrics may not properly capture the dissimilarities in the structural (spectral) properties of the graphs.

# 4 COMPUTING GGDs BETWEEN GRAPHS OF THE SAME SIZE

## 4.1 PHASE 1: SPECTRAL GRAPH MATCHING FOR FINDING NODE CORRESPONDENCE

Computing the GGD metric between two input graphs requires solving a graph-matching problem in advance. Without knowing the node-to-node correspondence that can be achieved through a graph-matching step, the distance between modified Laplacian matrices may be significantly higher than the minimum possible distance. In this work, we aim to find the infimum between two SPD matrices on the Riemannian manifold formed by modified Laplacian matrices, which can be accomplished through a graph-matching phase. Graph matching techniques can be used to establish node-to-node correspondence by seeking a bijection between node sets to maximize the alignment of edge sets (Livi & Rizzi, 2013; Emmert-Streib et al., 2016; Caetano et al., 2009). This combinatorial optimization problem can be cast into a Quadratic Assignment Problem, which is NP-hard to solve or approximate (Fan et al., 2020; Wang et al., 2020).

In this study, we will exploit a spectral graph matching method called GRAMPA (**GRA**ph **M**atching by **P**airwise eigen-**A**lignments) (Fan et al., 2020) to find the approximate node correspondence between two graphs. GRAMPA starts with comparing the eigenvectors of the adjacency matrices of the input graphs. Instead of comparing only the eigenvectors responding to the largest eigenvalues, it considers all pairs

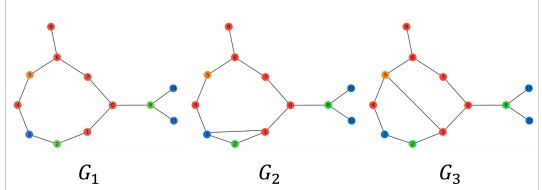

Figure 1: Graphs with simple perturbations.

of eigenvectors/eigenvalues to generate a similarity matrix. This similarity matrix can be constructed by summing up the outer products of eigenvector pairs, weighted by a Cauchy kernel (Fan et al., 2020). Subsequently, a rounding procedure will be performed to determine the optimal match between nodes employing the similarity matrix.

**Definition 4.1** (Similarity Matrix). Let $G_1$ and $G_2$ be two undirected graphs with $n$ nodes, and let their weighted adjacency matrices be $A_1$ and $A_2$, respectively. The spectral decompositions of $A_1$ and $A_2$ are expressed as follows: $A_1 = \sum_{i=1}^{n} \zeta_i u_i u_i^\top$ and $A_2 = \sum_{j=1}^{n} \mu_j v_j v_j^\top$, where the eigenvalues are ordered such that $\zeta_1 \geq \ldots \geq \zeta_n$ and $\mu_1 \geq \ldots \geq \mu_n$. The similarity matrix $\widehat{X} \in \mathbb{R}^{n \times n}$ is defined as:

$$\widehat{X} = \sum_{i,j=1}^{n} w\left(\zeta_i, \mu_j\right) \cdot u_i u_i^\top \mathbf{J} v_j v_j^\top, \text{where } w(x,y) = \frac{1}{(x-y)^2 + \eta^2}. \tag{6}$$

Here, $\mathbf{J} \in \mathbb{R}^{n \times n}$ denotes an all-one matrix and $w$ is the Cauchy kernel of bandwidth $\eta$.

The permutation estimate matrix $\widehat{\pi}$ can be obtained by rounding $\widehat{X}$, typically achieved by solving the Linear Assignment Problem (LAP):

$$\widehat{\pi} = \operatorname{argmax} \sum_{i=1}^{n} \widehat{X}_{i,\pi(i)}, \tag{7}$$

which can be efficiently solved using the Hungarian algorithm (Fan et al., 2020). However, one simpler rounding procedure was advocated in (Fan et al., 2020) with theoretical results supporting the rounding procedure, which is given by the following equation:

$$\widehat{\pi}(i) = \operatorname{argmax}_{j} \widehat{X}_{ij}, \tag{8}$$

here the permutation estimate matrix is constructed by selecting the largest index from each row. While LAP provides optimal matching, its computational complexity can become expensive for very large graphs. By carefully choosing $\eta$, the same match recovery holds if rounding is performed using equation 7 instead of solving the LAP in equation 8 (Fan et al., 2020).

**Lemma 4.1** (Graph Matching Recovery). *Given symmetric matrices $A_1$, $A_2$ and $Z$ from the Gaussian Wigner model, where $A_{2\pi^*} = A_1 + \sigma Z$, there exist constants $c, c' > 0$ such that if $1/n^{0.1} \leq \eta \leq c/\log n$ and $\sigma \leq c'\eta$, then with probability at least $1 - n^{-4}$, GRAMPA Algorithm correctly recovers the permutation matrix $\pi^*$ from the Similarity matrix $\widehat{X}$ (Fan et al., 2020). Its proof can be found in the supporting documents A.6.*

Once $\widehat{\pi}$ is obtained, the best-matched mirrors of the input graphs are:

$$\text{Best Match to } A_2 = \widehat{\pi} A_1 \widehat{\pi}^\top, \quad \text{Best Match to } A_1 = \widehat{\pi}^\top A_2 \widehat{\pi}. \tag{9}$$

In practice, the graph matching performance is not too sensitive to the choice of tuning parameter $\eta$. For small-sized graphs, such as the MUTAG dataset(Morris et al., 2020), setting $\eta = 0.5$ yields satisfactory results in matching. In Appendix A.11, the effect of $\eta$ for computing GGDs has been comprehensively analyzed.

## 4.2 PHASE 2: COMPUTING GEODESIC DISTANCES BETWEEN GRAPH LAPLACIANS

The GGD metric can be formally defined as the infimum length of geodesics connecting two data points in the Riemannian manifold formed by the cone of the modified graph Laplacian matrices (Lim et al., 2019). This distance metric can be imagined as a matrix representation of the geometric distance $|\log(a/b)|$ between two positive numbers $a, b$ (Bonnabel & Sepulchre, 2010; Shamai & Kimmel, 2017; Owen & Provan, 2010).

**Definition 4.2** (Graph Geodesic Distance). Let $\mathcal{L}_1$ and $\mathcal{L}_2 \in \mathbb{S}_{++}^n$ denote two modified Laplacian matrices corresponding to two matched graphs $G_1$ and $G_2$ both having $n$ nodes, then their Graph Geodesic Distance denoted by $GGD(G_1, G_2) : \mathbb{S}_{++}^n \times \mathbb{S}_{++}^n \to \mathbb{R}_+$, is defined as:

$$GGD(G_1, G_2) = \left[ \sum_{i=1}^{n} \log^2(\lambda_i(\mathcal{L}_1^{-1} \mathcal{L}_2)) \right]^{1/2}, \tag{10}$$

where $\lambda_i$ are the generalized eigenvalues computed with the matrix pencil $(\mathcal{L}_1, \mathcal{L}_2)$.

The above GGD formulation for computing distances between SPD matrices is based on an Affine-Invariant Riemannian Metric (AIRM) (Lim et al., 2019), while another well-known metric, the Log-Euclidean Riemannian Metric (LERM) (Ilea et al., 2018; Thanwerdas & Pennec, 2023; Chen et al., 2024) is also discussed in Appendix A.9 of the supplementary section.

## 4.3 CONNECTION BETWEEN GGD AND GRAPH STRUCTURAL MISMATCHES

Consider two graphs, $G_1$ and $G_2$, that have the same node set $V$, with a known correspondence between their nodes. Let $L_1$ and $L_2$ be the Laplacian matrices of these graphs, respectively. Suppose we take a subset of nodes, denoted by $S$ and its complement, $S'$. We assign the value 1 to the nodes in $S$ and the value 0 to those in $S'$. This defines the set $S$ as:

$$S \overset{\text{def}}{=} \{v \in V : x(v) = 1\}.$$

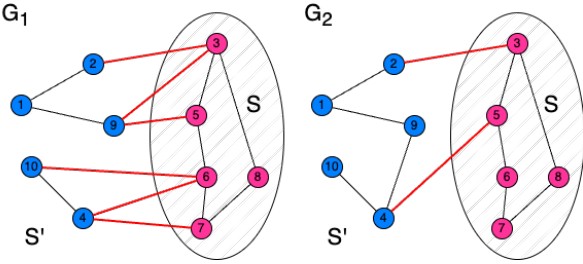

For graph $G_1$, the cut for the node subset $S$ (which is the number of edges that cross between $S$ and $S'$) can be computed as:

$$\text{cut}_{G_1}(S, S') = x^T L_1 x.$$

As shown in Figure 2, for node subset $S$ six edges cross between $S$ and $S'$ in graph $G_1$, whereas only two edges cross in graph $G_2$. This difference in edge counts between the two graphs is referred to as a cut mismatch. The relationship between

Figure 2: The cut mismatch (for the node set $S$) between two simple graphs is $\frac{6}{2} = 3$.

this cut mismatch and the generalized eigenvalue problem for the matrix pair $(L_1, L_2)$ can be formalized using the Generalized Courant-Fischer Minimax Theorem (Golub & Van Loan, 2013; Feng, 2020).

**Lemma 4.2** (The Generalized Courant-Fischer Minimax Theorem). *Given two Laplacian matrices* $L_1, L_2 \in \mathbb{R}^{n \times n}$ *such that* $\text{null}(L_2) \subseteq \text{null}(L_1)$, *the $k$-th largest generalized eigenvalue of $L_1$ and $L_2$ can be computed as follows for* $1 \leq k \leq \text{rank}(L_2)$:

$$\lambda_k = \min_{\substack{\dim(U)=k \\ U \perp \text{null}(L_2)}} \max_{x \in U} \frac{x^\top L_1 x}{x^\top L_2 x}. \tag{11}$$

This theorem provides a method to bound the maximum cut mismatch between two graphs by calculating the largest generalized eigenvalue. Specifically, we can use the following optimization problem to compute the dominant eigenvalue $\lambda_{max}$ (Feng, 2020):

$$\lambda_{\max} = \max_{\substack{|x| \neq 0 \\ x^\top \mathbb{1} = 0}} \frac{x^\top L_1 x}{x^\top L_2 x} \geq \max_{\substack{|x| \neq 0 \\ x(v) \in \{0,1\}}} \frac{x^\top L_1 x}{x^\top L_2 x} = \max \frac{cut_{G_1}(S, S')}{cut_{G_2}(S, S')}. \tag{12}$$

From equation (12), we can see that the dominant generalized eigenvalue $\lambda_{max}$ corresponds to the most significant cut mismatch between $G_1$ and $G_2$. In particular, $\lambda_1 = \lambda_{max}$ sets an upper bound on the cut mismatch between $G_1$ and $G_2$, while $\lambda_n = \lambda_{min}$ defines the upper bound of the mismatch in the reverse direction, between $G_2$ and $G_1$. Appendix A.7 illustrates this relationship with practical examples.

## 5 COMPUTING GGDs FOR GRAPHS WITH DIFFERENT SIZES

**Submatrix selection methods** To calculate geodesic distances between SPD matrices of different sizes, prior studies have proposed a submatrix adaptation method (Lim et al., 2019). In this approach, a principle submatrix with the same size as the smaller matrix is obtained from the larger matrix (Ye & Lim, 2016), and subsequently used to calculate the GGD. Furthermore, this method can be extended to project the smaller matrix into a larger one with the same size as the larger matrix (Lim et al., 2019). While these methods are efficient for handling SPD matrices, for our application taking the submatrix of the modified Laplacian can lose important nodes/edges, compromising critical graph structural properties.

**Graph coarsening methods** In this work, we will leverage spectral graph coarsening to address the issue. Spectral graph coarsening is a widely adopted process (Loukas, 2019; Aghdaei & Feng, 2022) for reducing graph sizes while preserving key spectral (structural) properties, such as the Laplacian eigenvalues/eigenvectors. Recent spectral graph coarsening methods aim to decompose an input graph into many distinct node clusters, so that a reduced graph can be formed by treating each node cluster as a new node, with a goal of assuring that the reduced graph will approximately retain the original graph's structure (Loukas, 2019; Han et al., 2024; Aghdaei & Feng, 2022). Therefore, when computing GGDs for graphs of different sizes, we can first adopt spectral graph coarsening to transform the bigger graph into a smaller one, so that our framework in Section 4 can be subsequently utilized. However, existing state-of-the-art graph coarsening methods do not allow us to precisely control the size of the reduced graphs.

## 5.1 Our Approach: Spectral Graph Coarsening by Effective Resistances

In this work, we introduce a spectral graph coarsening method using effective-resistance clustering (Aghdaei & Feng, 2022). Our approach starts with estimating the effective resistances of all edges in the original graph. We can also incorporate the difference between node features (if available) as an additional parameter. In the graph coarsening phase, our method will rank edges according to their resistance distances and only the top few edges with the smallest resistances will be coarsened into new nodes. This approach enables precise control over the size of the reduced graphs while preserving crucial structural properties, such as the eigenvalues/eigenvectors of the adjacency matrices, which are essential for the subsequent spectral graph matching step (Phase 1 in Section 4.1).

Consider a connected, weighted, undirected graph $G = (V, E, w)$ with $|V| = n$. The effective resistance between nodes $(p, q) \in V$ plays a crucial role in various graph analysis tasks including spectral sparsification algorithms (Spielman & Teng, 2011). The effective resistance distances can be accurately computed using the equation:

$$R_{eff}(p, q) = \sum_{i=2}^{n} \frac{(u_i^\top b_{pq})^2}{\sigma_i}, \tag{13}$$

where $b_p \in \mathbb{R}^n$ denote the standard basis vector with all zero entries except for the $p$-th entry being 1, and $b_{pq} = b_p - b_q$. $u_i \in \mathbb{R}^n$ for $i = 1, \ldots, n$ denote the unit-length, mutually-orthogonal eigenvectors corresponding to Laplacian eigenvalues $\sigma_i$ for $i = 1, \ldots, n$. A brief background on effective resistance is provided in Appendix A.4.

**Scalable estimation of effective resistances** To address the computational complexity associated with directly computing eigenvalues and eigenvectors required for estimating edge effective resistances, we leverage a scalable framework for approximating the eigenvectors of the graph Laplacian matrix using the Krylov subspace (Saad, 2011). Let $A$ denote the adjacency matrix of a graph $G$, consider its order-$m$ Krylov subspace $\mathbf{K}_m(A, x)$ that is a vector space spanned by the vectors computed through power iterations $x, Ax, A^2x, \ldots, A^{m-1}x$ (Liesen & Strakoš, 2012). By enforcing orthogonality among the above vectors in the Krylov subspace, a new set of mutually orthogonal vectors of unit lengths can be constructed for approximating the original Laplacian eigenvectors in 13, which are denoted as $\tilde{u}_1, \tilde{u}_2, \ldots, \tilde{u}_m$. To estimate the effective resistance between two nodes $p$ and $q$, we can exploit the approximated eigenvectors:

$$R_{eff}(p, q) \approx \sum_{i=1}^{m} \frac{(\tilde{u}_i^\top b_{pq})^2}{\tilde{u}_i^\top L \tilde{u}_i}, \tag{14}$$

where $\tilde{u}_i$ represents the approximated eigenvector corresponding to the $i$-th eigenvalue of $L$.

**Graph coarsening with node features** In order to account for the variation in node features along with edge resistive distance, we can use the following modified effective resistance formulation:

$$R_{eff}^*(p, q) = R_{eff}(p, q) + \alpha \| f_p - f_q \|, \tag{15}$$

where $f_p$ and $f_q$ are node feature vectors of nodes $p$ and $q$, respectively, while $\alpha$ is a weighting factor that determines the effect of node feature information in the graph coarsening process. For instance, if the weight is sufficiently large, the modified effective resistance between nodes with different features will always exceed that of nodes with similar features, effectively preventing their coarsening.

## 6 GGD as a Distance Metric

Assuming the graph matching problem can always find the exact correspondence between nodes, we prove that the GGD metric (based on AIRM) between any two nonempty graphs is a metric that satisfies the following conditions:

- The distance between a graph and itself or between two isomorphic graphs is zero: $GGD(G, G) = 0$.
- (Positivity) The distance between two distinct graphs is positive: $GGD(G_1, G_2) \geq 0$.
- (Symmetry) The distance between $G_1$ and $G_2$ is the same of the one between $G_2$ and $G_1$: $GGD(G_1, G_2) = GGD(G_2, G_1)$.
- The triangle inequality: $GGD(G_1, G_3) \leq GGD(G_1, G_2) + GGD(G_2, G_3)$.

Detailed proofs of the above four properties are provided in Appendix A.5.

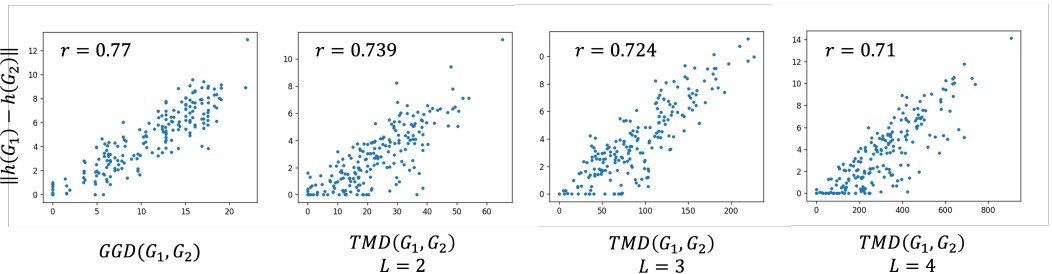

Figure 3: Correlation between graph distance metrics and GNN model outputs.

# 7 EXPERIMENTS

## 7.1 APPLICATION OF GGDs IN GNN STABILITY ANALYSIS

To analyze the stability of GNN models (Bronstein et al., 2017; Garg et al., 2020; Duvenaud et al., 2015), we conducted multiple experiments with the GGD and TMD metrics. GNNs typically operate by a message-passing mechanism (Gilmer et al., 2017), where at each layer, nodes send their feature representations to their neighbors. The feature representation of each node is initialized to its original features and is updated by repeatedly aggregating incoming messages from neighbors. In our experiment, we relate GGD to the Graph Isomorphism Networks (GIN) (Xu et al., 2019), one of the most widely applied and powerful GNNs, utilizing the MUTAG dataset (Morris et al., 2020) as our reference graph dataset. The objective is to analyze the relationship between the input distance $GGD(G_1, G_2)$ and the distance between the output GIN vectors, $\|h(G_1) - h(G_2)\|$ for randomly selected pairs of graphs.

We employed a three-layer GIN network as described in (Xu et al., 2019). This network uses GIN convolutional layers to update tensors of nodes based on their neighboring nodes and then aggregates those outputs in a vector representation, followed by linear layers for classification tasks. Thus it outputs a single vector $h(G)$ for the entire graph $G$. The result is illustrated in Figure 3.

We observe a strong correlation between GGD and the output distance, as indicated by a high Pearson correlation coefficient. This finding implies the effectiveness of the proposed GGD metric for analyzing the stability of GNN models (Chuang & Jegelka, 2022). To compare GGD with existing metrics, we repeat this experiment using TMD without considering node attributes (features). As shown in Figure 3, GGD demonstrates a better correlation with GIN outputs than the TMD metric across different levels. These findings indicate that when dealing with graphs without node features, GGD should be adopted for the stability analysis of graph learning models.

## 7.2 APPLICATION OF GGDs IN GRAPH CLASSIFICATION TASKS

We evaluate whether the GGD metric aligns with graph labels in graph classification tasks using datasets from TUDatasets (Morris et al., 2020). We employ a Support Vector Classifier (SVC) ($C = 1$) with an indefinite kernel $e^{-\gamma * GGD(G_1, G_2)}$, which can be viewed as a noisy observation of the true positive semidefinite kernel (Luss & d'Aspremont, 2007). The parameter $\gamma$ is selected through cross-validation from the set {0.01, 0.05, 0.1}. For comparative analysis with existing methods, we include graph kernels based on graph subtrees: the WL subtree kernel (Shervashidze et al., 2011); and two widely adopted GNNs: graph isomorphism network (GIN) (Xu et al., 2019) and graph convolutional networks (GCN) (Kipf & Welling, 2017).

Table 2 presents the mean and standard deviation over five independent trials with a 90%-10% train-test split. For most cases, GGD consistently outperforms the performance of state-of-the-art GNNs, graph kernels, and metrics when node attributes are missing. Additionally, we observe that GGD allows us to obtain better results for larger datasets than smaller ones.

Table 2: Classification accuracies for various models on graph datasets.

| Dataset | Accuracy in percentage | | | |
|---|---|---|---|---|
| | MUTAG | PC-3H | SW-620H | BZR |
| GGD | 86.24±7.89 | **78.34±1.60** | **77.6±3.50** | **83.23±6.25** |
| TMD, L = 2 | 76.19±5.26 | | | |
| TMD, L = 3 | 77.34±5.26 | 71.24±2.45 | 70.22±2.29 | 73.43±2.44 |
| TMD, L = 4 | 78.20±5.26 | 71.37±1.42 | 70.84±2.29 | 73.96±4.88 |
| TMD, L = 5 | | 71.89±2.40 | 71.20±1.88 | 75.13±2.44 |
| GCN(Kipf & Welling, 2017) | 77.37±3.95 | 70.56±1.66 | 69.44±0.94 | 72.56±3.66 |
| GIN(Xu et al., 2019) | 82.60±4.60 | 75.34±1.10 | 73.36±2.32 | 77.09±3.66 |
| WWL(Togninalli et al., 2019) | 72.39±2.63 | 65.46±1.11 | 68.06±0.86 | 72.37±1.22 |
| WL Subtree(Shervashidze et al., 2011) | 76.81±6.30 | 68.43±0.76 | 69.36±1.20 | N/A |
| FGW(Titouan et al., 2019) | **88.33±5.26** | 61.77±1.11 | 58.28±1.02 | 51.03±2.63 |

Table 3: Runtime comparison for different distance metrics on graph datasets.

| | MUTAG | PC-3H | SW-620H | BZR |
|---|---|---|---|---|
| GGD | **4.87 s** | **31.89 s** | **45.37 s** | **5.80 s** |
| TMD, L = 3 | 5.30 s | 88.60 s | 98.70 s | 7.22 s |
| TMD, L = 4 | 7.89 s | 111.98 s | 134.38 s | 10.34 s |
| TMD, L = 6 | 11.27 s | 272.50 s | 288.12 s | 14.98 s |

## 7.3 RUNTIME COMPLEXITY ANALYSIS AND COMPARISON

When comparing various graph distance metrics, a primary consideration is their computational complexity. Conventional approaches usually require intricate computations that frequently have cubic time or higher complexities. For our problem, the spectral graph matching step requires the eigenvalue decomposition of adjacency matrices and solving the linear assignment problem (LAP). Eigenvalue decomposition of an $n \times n$ matrix has a complexity of $O(n^3)$ (Borodin & Munro, 1975; Flamary et al., 2021), while solving the LAP using the Hungarian algorithm also has a runtime complexity of $O(n^3)$. Similarly, calculating the generalized eigenvalue of two SPD matrices entails a cubic complexity. Consequently, the overall complexity of GGD calculation is $O(n^3)$. On the other hand, TMD is an OT-based distance metric with a complexity of $O(n^3 \log(n))$ (Chuang & Jegelka, 2022; Flamary et al., 2021). Therefore, GGD exhibits slightly better (lower) runtime complexity than the TMD metric.

To evaluate runtime performance, we conduct extensive experiments to compare the runtime of computing TMD at various levels with GGD on both small graphs (MUTAG, BZR) and large graphs (PC-3H, SW-620H) collected from the TUDataset (Morris et al., 2020). Table 3 presents the average runtime (in seconds) for computing 100 distances between different graphs obtained by repeating the experiment five times. The results demonstrate that GGD consistently outperforms TMD in terms of runtime across all datasets and scenarios, particularly when dealing with larger graphs that contain more nodes. The TMD metric computation usually requires more levels to effectively capture the entire graph structure. In such cases, GGD exhibits runtime performance approximately $6 - 9$ times faster than of TMD. Hence, we conclude that GGD is significantly more computationally efficient than TMD, especially when working with large graphs. More details about our experimental setup can be found in Appendix A.12.

## 7.4 PARTIAL NODE FEATURES

Cutting-edge graph distance metrics like TMD, rely on node attributes to compute the dissimilarity between graphs, resulting in more accurate outcomes when all attributes are available. However,

Table 4: Comparison of correlation with GNN outputs and distance metrics with partial node features.

| Distance metric | Node features missing in percentage | | | | |
|---|---|---|---|---|---|
| | 0% | 20% | 50% | 80% | 100% |
| TMD, L = 3 | **0.84** | **0.78** | 0.72 | 0.63 | 0.61 |
| TMD, L = 4 | 0.81 | 0.77 | 0.62 | 0.58 | 0.57 |
| TMD, L = 5 | 0.80 | 0.75 | 0.65 | 0.58 | 0.53 |
| GGD | 0.78 | **0.78** | **0.77** | **0.77** | **0.77** |

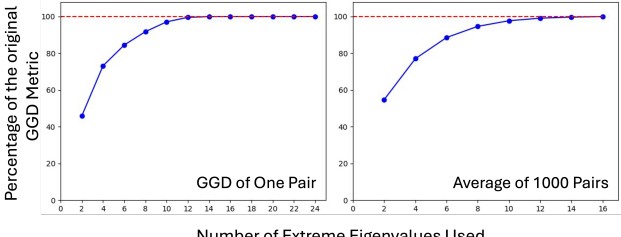

Figure 4: Percentage of the original GGD using numbers of extreme eigenvalues.

acquiring datasets with complete node attributes is often unattainable in real-world scenarios, leading to situations where certain features are partially missing. In such scenarios when only partial node features are available, we compare TMD with GGD to better understand their differences. Table 4 shows that the TMD metric outperforms GGD at various levels when node features are fully accessible. However, when node features are randomly removed from the MUTAG dataset, the accuracy of TMD degrades substantially.

### 7.5 GGD APPROXIMATION USING EXTREME EIGENVALUES

The largest and the smallest eigenvalues correspond to the most dominant mismatches in graph cuts and effective resistance distances, contributing the most to the total GGD value. Similarly, the second largest and smallest eigenvalues correspond to the next significant mismatched cuts. In our experiment, we obtain approximate GGDs using a few extreme eigenvalue pairs and compare them with the ground truth. Figure 4 illustrates the relative accuracy of the approximate GGDs, in which we observe that the top four pairs of extreme eigenvalues contribute $80\%$ of the total GGD values. In addition, we conduct the SVC classification task and GNN correlation study using GGD with only $2$ and $4$ extreme eigenvalue pairs, respectively, and present the associated findings in Table 5.

## 8 CONCLUSION

In this work, we introduce Graph Geodesic Distance (GGD), a novel spectral graph distance metric based on graph matching and the infimum on a Riemannian manifold. GGD captures the essential structural mismatches of graphs vital for graph classification tasks. Additionally, we show that GGD can serve as an effective metric for analyzing the stability of GNN models and graph classification tasks, achieving superior performance when only partial node features are available.

Table 5: Performance of GGD using extreme eigenvalues only.

| Task | Number of extreme eigenvalues | | |
|---|---|---|---|
| | 2 | 4 | All |
| Correlation with GNN | 0.74 | 0.76 | 0.77 |
| Classification accuracy | $81.50 \pm 6.85$ | $83.87 \pm 7.56$ | $86.00 \pm 7.50$ |

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

# A APPENDIX

## A.1 ALGORITHM FLOW

---

**Algorithm 1** GGD: Geodesic Graph Distance

---

1: **Input:** Graphs $G_1 = (V_1, E_1, w_1)$, $G_2 = (V_2, E_2, w_2)$, tuning parameter $\eta > 0$, small diagonal value $0 < \epsilon \ll 1$, node feature weight $\alpha$
2: **Output:** GGD Value
3: Compute the adjacency matrices $A_1$, $A_2$
4: **if** shape($A_1$) $\neq$ shape($A_2$) **then**
5:     Assign the larger graph to $G_1$, and the smaller graph to $G_2$
6:     **while** shape($A_1$) > shape($A_2$) **do**
7:         Compute the effective resistance $R_{eff}(p, q)$ of each edge $(p, q) \in E_1$
8:         Compute the modified effective resistance $R_{eff}^*(p, q) = R_{eff}(p, q) + \alpha||NF_p - NF_q||$
9:         Coarsen the edge with the lowest $R_{eff}^*(p, q)$
10:        Update $A_1$
11:     **end while**
12: **end if**
13: Compute eigenvectors $u_i$, $v_i$ and eigenvalues $\zeta_i$, $\mu_i$ of $A_1$ and $A_2$, respectively
14: Compute the similarity matrix $\hat{X} \in \mathbb{R}^{n \times n}$
15: Solve Linear Assignment Problem to compute the permutation estimate matrix $\hat{\pi}$
16: Update $A_2$ by multiplying with $\hat{\pi}$ to get best match with $A_1$
17: Derive $L_1$ and $L_2$ from $A_1$ and $A_2$
18: Add $\epsilon$ to diagonal values of $L_1$ and $L_2$
19: Compute GGD value using the generalized eigenvalues
20: **return** GGD

---

## A.2 GRAPH ADJACENCY AND LAPLACIAN MATRICES

For an undirected graph $G = (V, E, w)$, where $V$ represents the set of nodes (vertices), $E$ represents the set of edges, and $w$ denotes the associated edge weights, the adjacency matrix $A$ is defined as follows:

$$A(i, j) = \begin{cases} w(i, j), & \text{if } (i, j) \in E. \\ 0, & \text{otherwise.} \end{cases} \tag{16}$$

Let $D$ denote the diagonal matrix where $D(i, i)$ is equal to the (weighted) degree of node $i$. The graph Laplacian matrix is then given by $L = D - A$. The rank of the Laplacian matrix of a graph $G$ is $n - c(G)$, where $n$ is the number of nodes and $c(G)$ is the number of connected components in the graph. For a connected graph, this implies that the rank of the Laplacian matrix is $n - 1$, meaning Laplacian matrices are not full-rank (Bondy et al., 1976).

## A.3 RIEMANNIAN MANIFOLD

A manifold is a type of topological space that resembles Euclidean space in small, local regions around each point. In other words, for every point on a manifold, there is a neighborhood that is similar to a flat multidimensional space. A Riemannian manifold is a type of manifold equipped with a smoothly varying inner product on the tangent spaces at each point. This means that for every point on the manifold, the tangent space has a way of measuring distances and angles, and these measurements change smoothly from point to point (Lee, 2018). In simpler terms, a Riemannian manifold is a smooth, curved space that locally behaves like Euclidean space but has its own geometric properties, such as how distances, angles, and volumes are defined. These properties are determined by a Riemannian metric, which generalizes the concept of measuring lengths and angles in flat space to curved spaces (Lee, 2018).

A Riemannian manifold can have curvature, unlike a flat space. This curvature allows the study of geometric shapes ranging from spheres and cylinders to more abstract surfaces. The Riemannian structure enables us to compute geodesics, volumes, and various types of curvature. This makes Riemannian manifolds fundamental in fields like differential geometry and physics, and increasingly

important in data science, where curved spaces are used to model complex datasets (You & Park, 2021).

### A.4 Effective Resistance in Graph Theory

Effective resistance, also known as resistance distance, is a concept in graph theory that measures the electrical resistance between two nodes in a weighted or unweighted graph represented as a network of resistors (Ghosh et al., 2008). It draws an analogy between electrical networks and graphs, helping to quantify how easily current can flow between two nodes, where the edges are treated as resistors. The effective resistance between nodes provides insight into the connectivity between the network. This means two nodes with lower effective resistance values have higher connectivity (Ellens et al., 2011).

### A.5 Detailed Proofs Showing GGD is a Metric

#### A.5.1 Identity Property

*Proof.* Let the corresponding SPD matrix of the graph $G$ be $\mathcal{L} \in \mathbb{S}_{++}^n$. From Equation 10, we have:

$$GGD(G,G) = \left[ \sum_{i=1}^{n} \log^2(\lambda_i(\mathcal{L}^{-1}\mathcal{L})) \right]^{1/2} = \left[ \sum_{i=1}^{n} \log^2(\lambda_i(I)) \right]^{1/2}.$$

The identity matrix has only one eigenvalue, which is 1. So, $GGD(G,G) = \left[ \log^2(1) \right]^{1/2} = 0$. $\square$

#### A.5.2 Positivity Property

*Proof.* Let the corresponding SPD matrices of the graphs $G_1$ and $G_2$ be $\mathcal{L}_1, \mathcal{L}_2 \in \mathbb{S}_{++}^n$. Let the generalized eigenvalues of $(\mathcal{L}_1^{-1}\mathcal{L}_2)$ be $\lambda_1, \lambda_2, \lambda_3, \ldots, \lambda_n$. From Equation 10, we get:

$$GGD(G_1, G_2) = \left[ \log^2(\lambda_1) + \log^2(\lambda_2) + \log^2(\lambda_3) + \ldots + \log^2(\lambda_n) \right]^{1/2}.$$

Now, $\log^2(\lambda_1) + \log^2(\lambda_2) + \log^2(\lambda_3) + \ldots + \log^2(\lambda_n) \geq 0$, for any values of $\lambda_i$.

We can conclude, $GGD(G_1, G_2) \geq 0$. $\square$

#### A.5.3 Symmetry Property

*Proof.* Let the corresponding SPD matrices of the graphs $G_1$ and $G_2$ be $\mathcal{L}_1, \mathcal{L}_2 \in \mathbb{S}_{++}^n$. Let the generalized eigenvalues of $(\mathcal{L}_1^{-1}\mathcal{L}_2)$ be $\lambda_1, \lambda_2, \lambda_3, \ldots, \lambda_n$. From Equation 10, we get:

$$GGD(G_1, G_2) = \left[ \sum_{i=1}^{n} \log^2(\lambda_i) \right]^{1/2}.$$

Given that the inverse of a symmetric matrix is also symmetric, and the product of two symmetric matrices is symmetric, it follows that $(\mathcal{L}_1^{-1}\mathcal{L}_2)$ is a symmetric tensor. Furthermore, the eigenvalues of a symmetric matrix and the eigenvalues of its inverse matrix are inversely related.

So, the eigenvalues of $(\mathcal{L}_2^{-1}\mathcal{L}_1)$ will be $\frac{1}{\lambda_1}, \frac{1}{\lambda_2}, \frac{1}{\lambda_3}, \ldots, \frac{1}{\lambda_n}$.

$$GGD(G_2, G_1) = \left[ \sum_{i=1}^{n} \log^2(\frac{1}{\lambda_i}) \right]^{1/2}.$$

Now, $\log\left(\frac{1}{\lambda_i}\right) = -\log(\lambda_i)$; so, $\log^2\left(\frac{1}{\lambda_i}\right) = \log^2(\lambda_i)$.

So, we can conclude $GGD(G_1, G_2) = GGD(G_2, G_1)$. $\square$

### A.5.4 TRIANGLE INEQUALITY

*Proof.* Let, $\mathcal{L}_1, \mathcal{L}_2, \mathcal{L}_3 \in \mathbb{S}_{++}^n$ are three SPD matrices corresponding to graphs $G_1, G_2, G_3$.

Now, The Frobenius norm $\|X\|_F$ is the geodesic length at $d(\exp X, I) = \|X\|_F$ (Bonnabel & Sepulchre, 2010). Hence at identity, $d(\mathcal{L}, I) = \|\log \mathcal{L}\|_F$.

From (Bonnabel & Sepulchre, 2010; You & Park, 2021) we get,

$$GGD(G_1, G_2) = GGD\left(G_1^{-1/2} G_2 G_1^{-1/2}, I\right) = \left\|\log\left(\mathcal{L}_1^{-1/2} \mathcal{L}_2 \mathcal{L}_1^{-1/2}\right)\right\|_F = \left\|\log\left(\mathcal{L}_1^{-1} \mathcal{L}_2\right)\right\|_F. \tag{17}$$

We know,

$$\mathcal{L}_1^{-1} \mathcal{L}_3 = \mathcal{L}_1^{-1}(\mathcal{L}_2 \mathcal{L}_2^{-1})\mathcal{L}_3 = (\mathcal{L}_1^{-1} \mathcal{L}_2)(\mathcal{L}_2^{-1} \mathcal{L}_3).$$

Now using the Frobenius norm inequality, we get:

$$\|\mathcal{L}_1^{-1} \mathcal{L}_3\| = \|(\mathcal{L}_1^{-1} \mathcal{L}_2)(\mathcal{L}_2^{-1} \mathcal{L}_3)\| \le \|\mathcal{L}_1^{-1} \mathcal{L}_2\| \|\mathcal{L}_2^{-1} \mathcal{L}_3\|.$$

Now taking logarithms on both sides:

$$\|\log(\mathcal{L}_1^{-1} \mathcal{L}_3)\| \le \|\log(\mathcal{L}_1^{-1} \mathcal{L}_2)\| + \|\log(\mathcal{L}_2^{-1} \mathcal{L}_3)\|.$$

Using Equation 10, we conclude:

$$GGD(G_1, G_3) \le GGD(G_1, G_2) + GGD(G_2, G_3).$$

$\square$

### A.6 GRAPH MATCHING RECOVERY

Given symmetric matrices $A_1$, $A_2$ and $Z$ from the Gaussian Wigner model, where $A_{2\pi^*} = A_1 + \sigma Z$, there exist constants $c, c' > 0$ such that if $1/n^{0.1} \le \eta \le c/\log n$ and $\sigma \le c'\eta$, then with probability at least $1 - n^{-4}$ for all large n, the matrix $\widehat{X}$ in equation 6 satisfies,

$$\min_{i \in [n]} \widehat{X}_{i, \pi^*(i)} > \max_{i,j \in [n]: j \ne \pi^*(i)} \widehat{X}_{ij} \tag{18}$$

and hence, the GRAMPA algorithm correctly recovers the permutation estimation matrix $\pi^*$.

Now, this proof is divided into two parts:

**Lemma A.1** (Noiseless Setting Diagonal Dominance). *In a noiseless situation, means replacing $A_2$ with $A_1$, similarity matrix $\widehat{X}^*$ is defined as:*

$$\widehat{X}^* = \widehat{X}(A_1, A_1) = \sum_{i,j=1}^n \frac{u_i u_i^T \mathbf{J} u_j u_j^T}{(\zeta_i - \zeta_j)^2 + \eta^2}. \tag{19}$$

*For some constants $C, c > 0$, if $1/n^{0.1} < \eta < c/\log n$, then with probability at least $1 - 5n^{-5}$ for large $n$, it can be proved that the diagonal components of $\widehat{X}^*$ are dominant by showing (Fan et al., 2019):*

$$\min_{i \in [n]} (\widehat{X}^*)_{ii} > \frac{1}{3\eta^2} \tag{20}$$

*and*

$$\max_{i,j \in [n]: i \ne j} (\widehat{X}^*)_{ij} < C\left(\frac{\sqrt{\log n}}{\eta^{3/2}} + \frac{\log n}{\eta}\right). \tag{21}$$

**Lemma A.2** (Bounding the Noise Impact). *The difference between the similarity matrix $X$ in the presence of noise and the noiseless situation is bounded. If $\eta > 1/n^{0.1}$, then for a constant $C > 0$, with probability at least $1 - 2n^{-5}$ for large $n$, it can be shown (Fan et al., 2019):*

$$\max_{i,j \in [n]} |\widehat{X}_{ij} - (\widehat{X}^*)_{ij}| < C\sigma\left(\frac{1}{\eta^3} + \frac{\log n}{\eta^2}\left(1 + \frac{\sigma}{\eta}\right)\right). \tag{22}$$

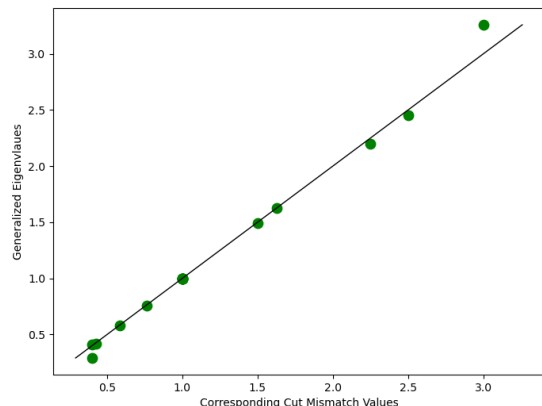

Figure 5: Generalized eigenvalues and their corresponding cut mismatches

Assuming lemma A.1 and A.2, for some $c, c' > 0$ sufficiently small, and by setting $\eta < c/\log n$ and $\sigma < c'\eta$, the algorithm ensures that the right sides of both equations 21 and 22 are at most $1/(12\eta^2)$. Then when $\pi^* = id$ (the identity permutation), these lemmas combine to imply:

$$\min_{i \in [n]} \widehat{X}_{ii} > \frac{1}{4\eta^2} > \frac{1}{6\eta^2} > \max_{i,j \in [n]: i \neq j} \widehat{X}_{ij} \tag{23}$$

with probability at least $1 - n^{-4}$. So, all diagonal entries of $\widehat{X}$ are larger than all off-diagonal entries, thereby achieving exact recovery (Fan et al., 2019).

## A.7 Relation between Generalized Eigenvalues with Cut Mismatch

We selected two graphs from the MUTAG dataset and computed their generalized eigenvalues following the procedure for calculating the Generalized Graph Distance (GGD), which involves determining the node-to-node correspondence. Subsequently, we considerd all possible subsets of nodes and evaluate their corresponding cut mismatches. As shown in Figure 5, each generalized eigenvalue is closely associated with a cut mismatch. This empirical observation supports our hypothesis that the GGD between two input graphs is strongly correlated with structural mismatches in graphs.

## A.8 Choosing $\epsilon$ for converting Laplacians to SPD matrices

Laplacian matrices are symmetric positive semi-definite (SPSD) matrices. To convert these to symmetric positive definite (SPD) matrices, we added a diagonal matrix with very small values ($\epsilon$). We used $0.0001$ as the small value ($\epsilon$) in our experiments. When working with Laplacian matrices of a weighted or unweighted graph, values significantly smaller than the edge weights of that graph have a minimal effect on the transformation. We conducted additional experiments with different small values and included the results in Tables 6 and 7. In Table 6, we observed that in our specific case with the MUTAG graph dataset (Morris et al., 2020), where all graphs are unweighted, any value less than $0.001$ has an almost negligible influence on the performance of the graph classification task. Additionally, when using values equal to or less than $0.001$, the GGD value remains almost the same, as shown in Table 7.

## A.9 Comparison of Two Different Riemannian Metrics for SPD Matrices

The two most commonly used Riemannian metrics on the SPD manifold are the Affine Invariant Riemannian Metric (AIRM) and the Log-Euclidean Riemannian Metric (LERM) (Ilea et al., 2018; Thanwerdas & Pennec, 2023; Chen et al., 2024). AIRM is a Riemannian metric that remains invariant

Table 6: Classification accuracy using MUTAG dataset with different values of $\epsilon$.

| Value of $\epsilon$ | Classification accuracy | Value of $\epsilon$ | Classification accuracy |
|---|---|---|---|
| 0.1 | $76.38 \pm 7.89$ | 1e-4 | $85.96 \pm 5.26$ |
| 5e-2 | $79.02 \pm 6.58$ | 1e-5 | $84.21 \pm 5.26$ |
| 1e-2 | $79.02 \pm 5.26$ | 1e-6 | $85.96 \pm 5.26$ |
| 5e-3 | $81.57 \pm 7.89$ | 1e-7 | $85.96 \pm 7.89$ |
| 1e-3 | $81.57 \pm 7.89$ | | |

Table 7: GGD values using MUTAG dataset for different values of $\epsilon$.

| Value of $\epsilon$ | Normalized GGD of a random graph pair (MUTAG[85], MUTAG[103]) | Average normalized GGD of 1000 pairs |
|---|---|---|
| 0.1 | 0.712 | 0.727 |
| 5e-2 | 0.827 | 0.834 |
| 1e-2 | 0.952 | 0.959 |
| 5e-3 | 0.978 | 0.979 |
| 1e-3 | 0.996 | 0.995 |
| 1e-4 | 0.9996 | 0.9995 |
| 1e-5 | 0.99995 | 0.99996 |
| 1e-6 | 0.999996 | 0.999996 |
| 1e-7 | 1 | 1 |

under affine transformations, meaning the metric is unaffected when matrices are transformed by any invertible operation. The geodesic distance between two SPD matrices, A and B, using AIRM is given by (You & Park, 2021; Moakher, 2005):

$$d_{\text{AIRM}}(A, B) = \| \log(A^{-1}B)\|_F = \left[ \sum_{i=1}^{n} \log^2(\lambda_i(A^{-1}B)) \right]^{1/2}. \tag{24}$$

On the other hand, LERM addresses some of the computational complexity challenges associated with AIRM by mapping SPD matrices to an Euclidean space through the matrix logarithm operation. In this Euclidean space, computations are simplified. The geodesic distance between two SPD matrices, A and B, using LERM is defined as (Huang et al., 2014):

$$d_{\text{LERM}}(A, B) = \| \log(A) - \log(B)\|_F. \tag{25}$$

In this paper, we primarily used AIRM to compute geodesics because of its stronger theoretical foundation and its ability to better explain graph cut mismatches. However, for comparison, we also conducted experiments using LERM. Figure 6 shows that the Graph Geodesic Distances computed using the LERM metric are highly correlated with those obtained using AIRM, though the GGD using AIRM demonstrates better performance overall. A detailed performance comparison of these two metrics is provided in Table 8.

### A.10 Using Normalized Laplacians for GGD Calculation

In many existing studies, the normalized Laplacian matrix is commonly used to study spectral graph properties (Chung, 1997). The normalized Laplacian matrix of a graph $G$ is defined as: $L_{norm} = I - A_{norm}$, where $A_{norm}$ is the normalized adjacency matrix. The normalized adjacency matrix is expressed as (Chung, 1997):

Table 8: Comparison between Riemannian metrics for GGD calculation.

| Riemannian metric | Correlation with GNN output | Classification accuracy |
|---|---|---|
| Affine-Invariant | **0.7786** | **86±7.5%** |
| Log-Euclidean | 0.7634 | 84.38% |

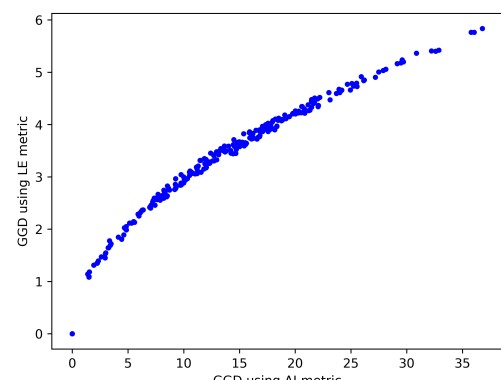

Figure 6: GGD between graph pairs using AI and LE Riemannian metric.

Table 9: Effect of epsilon in the calculation of GGD using normalized Laplacian matrices.

| Value of $\epsilon$ | 0.01 | 0.001 | 0.0001 | 0.00001 |
|---|---|---|---|---|
| GGD using Laplacian matrices | 16.235 | 16.775 | 16.832 | 16.838 |
| GGD using normalized Laplacian matrices | 384.097 | 254.440 | 188.345 | 165.332 |

$$A_{norm} = D^{-1/2}AD^{-1/2}, \tag{26}$$

where $D$ represents the diagonal degree matrix, and $A$ denotes the adjacency matrix of the graph.

Form equation 26, we can derive:

$$L_{norm} = I - D^{-1/2}AD^{-1/2} = D^{-1/2}(D - A)D^{-1/2} = D^{-1/2}LD^{-1/2}. \tag{27}$$

Similar to the Laplacian matrices of graphs, normalized graph Laplacian matrices are also symmetric and positive semi-definite. Therefore, it is necessary to add small values to the diagonal elements of these matrices. However, our experiments reveal that the GGD calculation is highly sensitive to this small value ($\epsilon$), resulting in significant fluctuations across different values, as demonstrated in Table 9. Additionally, the geodesic distances computed with the modified normalized Laplacian matrices exhibit poor accuracy in both classification tasks and their correlation with GNN outputs.

### A.11 EFFECT OF TUNING PARAMETER $\eta$ ON GRAPH MATCHING

In the original work, it was suggested that the regularization parameter $\eta$ needs to be chosen so that $\sigma \vee n^{-0.1} \lesssim \eta \lesssim 1/\log n$ (Fan et al., 2020). It is also mentioned that for practical cases, computing permutation matrix for different values of $\eta$ in an iterative way can result in better accuracy. The GRAMPA uses $\eta = 0.2$ for all their experiments (Fan et al., 2020).

We used a few values of $\eta$ in the classification problem using the MUTAG dataset and got that the best accuracy is obtained at $\eta = 0.5$. In Figure 7, the performance of the tuning parameter is demonstrated.

### A.12 EXPERIMENTAL SETUP

To evaluate the performance of the Graph Geodesic Distance (GGD) metric, we utilized graph datasets from the TUDataset collection (Morris et al., 2020). For small graphs, we used datasets like MUTAG and BZR, and for larger graphs, we selected PC-3H and SW-620H, which present more sizable networks. Detailed information about the datasets used is provided in Table 10.

While Classification tasks, each dataset was split into 90% training and 10% testing sets to ensure an unbiased evaluation process. When assessing the correlation with GNN, we trained a three-layer

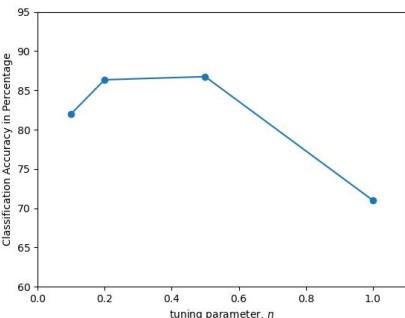

Figure 7: Classification accuracy vs GRAMPA tuning parameter.

Table 10: Brief description of graph datasets used.

| Dataset name | Number of graphs | Average number of nodes | Average number of edges |
|---|---|---|---|
| MUTAG | 188 | 17.93 | 19.79 |
| PC-3H | 27509 | 47.20 | 49.33 |
| SW-620H | 40532 | 26.06 | 28.09 |
| BZR | 405 | 35.75 | 38.36 |

GIN with 90% of all graphs from MUTAG and validated with the rest 10%. For the performance evaluation using graphs with partial node features, we took each dataset with node features and randomly removed a certain portion of features.

All experiments have been evaluated on a laptop with an Apple M1 chipset, featuring an eight-core CPU and a seven-core GPU.

