# OpenReview forum: "A Spectral Framework for Assessing the Geodesic Distance Between Graphs"
_ICLR.cc/2025/Conference — ICLR 2025 Conference Withdrawn Submission_

### Official Review · Reviewer_ZzaN · 2024-10-22

**Soundness:** 2
**Presentation:** 3
**Contribution:** 2
**Rating:** 5
**Confidence:** 4

**Summary:**

The paper introduces a new graph metric based on the geodesic distance between Laplacians on the manifold of positive definite symmetric matrices. The metric is used for tasks such as graph classification.

**Strengths:**

1. The metric introduced is demonstrated (Fig. 1 and Tab. 1) to be more powerful in capturing graph dissimilarities.
2. The authors have applied the metric to graph learning and showed its effectiveness.

**Weaknesses:**

1. My main concern is that the metric is not first introduced by the authors. It has been defined and thoroughly studied in [a]. Therefore, in this respect, the work is not considered novel and I don't think the contribution is enough for ICLR. However, some tricks introduced can be useful in practice.
2. Is it possible to incorporate a GGD-based SVC classifier with a GNN model?
3. The benchmarks (Tab. 2) are not recent.

[a] Lek-Heng Lim, Rodolphe Sepulchre, and Ke Ye. Geometric distance between positive definite matrices of different dimensions. IEEE Transactions on Information Theory, 65(9):5401–5405, 2019.

**Questions:**

Please see "Weaknesses".

---

> ### Author Response · Authors · 2024-11-24
>
> Answer to the weaknesses:
>
> 1. The Lim paper [1] does not address the problem of calculating a distance metric between graphs. Our work first performs graph matching and then converts the matched graphs to $n \times n$ real positive symmetric definite (SPD) matrices that naturally form a Riemannian manifold. Then the geodesic distance between data points is calculated using a Riemannian metric. Here graph matching is a crucial step that confirms the infimum distance between data points in the manifold. In the case of graphs of different sizes, we used a resistive distance-based spectral coarsening method to match their sizes and then proceeded with subsequent steps. To our knowledge, no previous work has used this method to determine the distance between graphs. The mentioned paper has influenced our idea generation and literature review, which we have properly cited in our paper. However, our problem statement is fundamentally different.
>
> 2. Common GNN models, such as Graph Isomorphism Networks (GIN), use message passing between convolutional layers, followed by one or more linear layers for the classification tasks. In contrast, Support Vector Classifiers (SVC) employ indefinite kernels that leverage the distances between points to distinguish between graph classes. To our knowledge, no method currently combines SVC with GNN classifiers. However, this could be an interesting direction for future work to improve classifier performance.
>
> 3. Thank you for bringing this to our attention. We have compared the accuracy of our classifier with the most common and widely used graph classification methods. Since our method does not require node features, we evaluated these classification models using graph datasets without node features to ensure a fair comparison. Although we are aware of some recent classification models that perform slightly better, we were unable to obtain their code, which is essential for us to benchmark on datasets without node features.
>
> [1] Lek-Heng Lim, Rodolphe Sepulchre, and Ke Ye. “Geometric distance between positive definite matrices of different dimensions.” IEEE Transactions on Information Theory, 65(9):5401–5405, 2019.

---

> > ### Comment · Reviewer_ZzaN · 2024-11-24
> >
> > Thank you for the replies. However, the foundation of the work is still GGD introduced by Lim et. al. Using spectral graph coarsening (a tool that is also not a contribution of the paper) is merely a technical step to address an issue that arises from computing GGD between matrices of different sizes. Therefore, I do not think the work is novel enough.

---

> > > ### Author Response · Authors · 2024-11-27
> > >
> > > In our work, we propose a multi-step approach. In the final phase, we compute the distance between SPD matrices in the Riemannian cone using the Riemannian metric, parallel to the work of Lim et al. A key contribution of our work is converting graph data into SPD matrices, a particularly challenging task. We address this through graph matching and the transformation of Laplacian matrices. Graph matching is crucial, as it ensures node-to-node correspondence, enabling the shortest possible distance on the manifold.
> > >
> > > We want to repeat a situation we mentioned for other reviewers. If we consider an example of two identical graphs with different node indices and calculate GGD without graph matching, they will produce different adjacency matrices and, subsequently, different SPD matrices. As a result, the distance will be non-zero. However, after performing the graph matching step, the matrices will align, and the distance will correctly be 0, as expected.

---

> > > > ### Comment · Reviewer_ZzaN · 2024-11-27
> > > >
> > > > Graph matching is also based on an existing work.

---

### Official Review · Reviewer_vjhM · 2024-10-30

**Soundness:** 3
**Presentation:** 3
**Contribution:** 3
**Rating:** 8
**Confidence:** 4

**Summary:**

The paper introduces a new graph distance known as GGD, which relies on a Laplacian based geodesic distance. In the case of comparing same dimensional matrices, an eigenvalue problem setup is adopted. In the case of comparing matrices of different dimensions, a effective resistance based coarsening scheme is adopted. The authors show theoretical properties (the distance capturing structural properties) and conducted experiments comparing with other distances.

**Strengths:**

originality: the proposed approach is, to the best of my knowledge, original.
quality: the quality of the exposition is good. The mathematical derivations (I have not checked proofs in supplement line by line) are, to my best understanding, reasonable and sound. The motivation is provided, and ample experiments/comparisons are made. Theoretical statements are relevant to the bigger picture.
clarity: the paper is written and presented clearly.
significance: the problem of graph metric development and graph comparison is significant, and the authors' contribution is original and considerable.

**Weaknesses:**

Overall, I believe the paper is written well and the contributions are sound and original.

- while the method is developed for general metric between graphs/matrices of different dimensions, the analysis in 4.3 only consider the case of same node set with known correspondence. The general case is not considered. Is there an analogous result for the general case with different dimensions, or with node features? The statement in the introduction/abstract makes it sound like the authors have section 4.3 type structural interpretation results for general case...if no such results are available, then the wording/claim should be specified/scaled down in the intro/abstract to reflect this more accurately.

- It is unclear whether the assumption in lemma 4.2 of null(L_2) \subseteq null(L_1) is a satisfiable assumption for general graphs in practice...it seems like a very strong assumption (beyond the trivial case that the constant vector is in the null space of laplacians for connected graphs...is this what the authors are trying to get at? ) Can the authors provide examples of common graph types where this assumption holds beyond the trivial case of connected graphs? How restrictive is this assumption in practice, and how does it impact the applicability of the method to real-world graphs?

**Questions:**

I state my questions below:

The authors introduce a very general framework for comparing graphs. I do not understand why they frame the abstract and the entire motivation of their paper around graph neural networks (as important as this application might be)...indeed, graph neural networks only enter as an application in the last sections of the paper...I find this emphasis in the abstract confusing. Consider revising the title and abstract to better reflect the general nature of the graph comparison framework and to more accurately represent the broader contributions of the paper.

---

> ### Author Response · Authors · 2024-11-24
>
> Answer to the weaknesses:
>
> 1. Thanks to the reviewer for pointing out this very relevant concern. In Section 4.3, we aim to explore the relationship between graph cut mismatches and the generalized eigenvalues of the modified Laplacian (SPD) matrices. Unfortunately, we do not have a rigorous mathematical proof that directly connects these quantities. Instead, we provide approximate relationships between them. The generalized eigenvalues are used to compute the distance value (via the AIRM method), and there is a well-established connection between eigenvalues and graph cuts mismatches. To simplify the analysis, we focus on graphs of the same size to reduce computational complexity.
>
> 2. Thank you for your comment. The only assumption is that both graphs are connected to use our framework for computing GGDs. Therefore, this assumption does not limit the applicability of the method in real-world scenarios.
>
> Answer to the questions:
>
> 1. Thanks to the reviewer for raising this question. We agree and have revised the abstract of our work to emphasize the development of the distance metric, with stability analysis and graph classification being crucial applications of our contribution.

---

### Official Review · Reviewer_rahK · 2024-11-03

**Soundness:** 1
**Presentation:** 1
**Contribution:** 2
**Rating:** 3
**Confidence:** 3

**Summary:**

This paper proposes new metrics to measure two different graphs (GGD) and apply them to graph neural networks.
The GGD exploits the spectral features of two graphs if the size of the two graphs is the same. If sizes are different, GGD exploits resistance.
This paper conducts experiments to confirm that the proposed method is more effective than the other existing measures.

**Strengths:**

The GGD empirically outperforms the existing measures.

**Weaknesses:**

The proof for GGD as a distance metric is incomplete. Would GGD give zero for the same graphs but two different adjacency matrices? The proof seems to assume that "phase 1" is already done, but would "phase 1" give the same modified matrix for two different matrices? This is not about making a fuss over details; this paper needs to provide more careful theoretical analyses than the current state by giving examples.

Section 4.3, where the authors discuss the connection between GGD and the cut, is rather weak. How does it support the GGD? How does the bound of the cut ratio relate to the quality of measure of the two graphs? From the current state, these are very unclear.

It is unclear The role of the modified Laplacian matrix; I speculate that this is to obtain the inverse of the graph Laplacian, but it seems that the pseudoinverse of Laplacian is enough. Can you please explain the role of the role of modification, especially the advantage over the pseudoinverse of the graph Laplacian?

**Questions:**

Connected to the first weakness point, if the resistances are the same, are the two graphs the same? If not, you may have a disadvantage for such cases if you employ the resistance.

Why do you use the approximated method for the resistance-based one? If two graphs are the same, the authors use all of the eigenvectors. The resistance costs almost the same as obtaining the full eigenvectors, but for this case, the authors approximate the resistance. Why?

Suggestion: Consider sorting all the proposed steps in the algorithm environment.

---

> ### Author Response · Authors · 2024-11-24
>
> Answer to the weaknesses:
>
> 1. As discussed in our work, Phase 1 involves finding the correspondence between nodes using a Graph Matching Technique. When we consider two different adjacency matrices of the same graph, the GRAMPA method identifies the permutation matrix that enables us to determine the approximate node-to-node correspondence. The proof for this scenario, where we account for noise (i.e., two different adjacency matrices), is provided in Lemma A.2 and is explained in detail in the GRAMPA paper [1]. Our empirical results demonstrated exact recovery in this case, meaning that after graph matching, the second adjacency matrix matches the first, provided they originate from the same graph.
>
> 2. In Section 4.3, we explained the relationship between GGD and the structural mismatch of graphs. After obtaining a node-to-node correspondence using a graph-matching algorithm, we can determine the graph cut mismatch between two graphs for a set of nodes. This mismatch is the ratio of edges going out of that set of graph nodes. In Equation (11), we show that the largest generalized eigenvalue of the Laplacian matrices of the graphs serves as the upper bound for the maximum possible graph cut mismatch between two graphs for any set of nodes S ⊆ V, where V is the set of all nodes. Similarly, the other eigenvalues quantify the graph cut mismatches between the input graphs. When using the generalized eigenvalues to calculate the GGD value in Equation (9), this value correlates with the graph cut mismatches of the two graphs. These graph cut mismatch ratios are closely related to the structural mismatch of the input graphs.
>
> 3. We used modified Laplacian matrices, which are Symmetric Positive Definite (SPD), due to their theoretical support in forming a natural Riemannian manifold.  Laplacian matrices are not Symmetric Positive Definite but are rather semi-definite. In the mathematical equation, it is clear that we need SPD matrices to compute their inverse for calculating the AIRM (Affine Invariant Riemannian Metric). However, this equation is used to calculate geodesics in the SPD manifold. Without the modification, using just the Laplacian matrices, we do not have proper information on whether the n*n Symmetric Positive Semi-Definite matrices create a Natural Riemannian Manifold or not. We have not found any previous work on this.
>
> Nevertheless, we conducted experiments using the Laplacian matrices directly to calculate the GGD through the pseudoinverse and observed very poor results. The accuracy for SVC using GGD with the pseudoinverse was around 56.88%, compared to our result of 86.24%.
>
> Answer to the questions:
>
> 1. In our method, we do not use resistances to determine whether the graphs are the same or not. Rather, if the graphs are the same or even of the same size, we do not need resistances. When the graphs are of different sizes, we calculate the resistive distances between the nodes of each edge in the larger graph and use coarsening until the sizes of the graphs are matched.
>
> 2. In the coarsening method, we need to calculate the effective resistance between nodes for all edges and repeat this process until we reach the target node count. Since this involves computing a large number of effective resistances, using the approximated method helps reduce computational cost. Furthermore, this approximation does not compromise our accuracy during the coarsening step. When the graphs reach the same size in terms of node numbers, we only need to calculate the eigenvalues once to obtain the similarity matrix.
>
> 3. Thanks to the reviewer for this valuable suggestion. In the supplementary section of our revised paper (Appendix A.1), we have included an algorithm flow.
>
> [1] Zhou Fan, Cheng Mao, Yihong Wu, and Jiaming Xu. “Spectral graph matching and regularized quadratic relaxations: Algorithm and theory”. In International conference on machine learning, pp. 2985–2995. PMLR, 2020.

---

> > ### Comment · Reviewer_rahK · 2024-11-27
> >
> > On the pseudo inverse point, I do agree with reviewer Z8Nj; there is a deep connection between PSD Laplacian and resistance. I still do not understand that what is the true impact of employing PD instead of PSD. I also agree with the other reviewers that this work is too incremental over Lim+. Thus, I maintain my score.

---

> > > ### Author Response · Authors · 2024-11-27
> > >
> > > We have addressed the concern raised by reviewer Z8Nj. In our work, there is no connection between SPD matrices (the modified Laplacian matrices) and effective resistance. Effective resistance is used solely for graph coarsening only when the graphs have different sizes. If two graphs have the same number of nodes, the coarsening step and therefore the effective resistance calculation is unnecessary. The SPD matrices are utilized only in the final stage, where we compute the distance in the Riemannian cone. Notably, effective resistance plays no role in the distance calculation phase, and SPD matrices are not required during the coarsening step.
> > >
> > > We also addressed the concern regarding the novelty of our work. Our approach involves multiple phases. In the final phase, we calculate the distance between SPD matrices in the Riemannian cone using Riemannian metrics, which is parallel to the work of Lim et al. However, converting graphs into the desired SPD matrices involves several challenging steps. Graph matching is a critical phase, as we have already mentioned. Without graph matching, we may obtain high distance values even for identical graphs with different node indices.

---

### Official Review · Reviewer_onsy · 2024-11-04

**Soundness:** 3
**Presentation:** 2
**Contribution:** 2
**Rating:** 3
**Confidence:** 3

**Summary:**

In this work, the authors introduce a new distance between graphs (denoted GGD). This distance differs from existing distances in that it makes use of the metric of positive definite matrices, in particular the Laplacians of graphs (after regularisation of the diagonal with the addition of $\eps Id$ to make it positive), to calculate a distance between two graphs with the same number of nodes. To do this, it is first needed to solve a correspondence problem between the nodes of the two graphs; the authors simply use the method of Fan et al. from 2020 (GRAMPA). When the graphs do not have the same size, the proposed method is to reduce the size of the larger one on spectral criteria (to keep the spectrum properties of the Laplacian of the reduced graph as close as possible to the spectrum of the initial graph). More precisely, the authors choose to compress the edges with the greatest effective resistance in the graph (which puts 2 nodes together) and thus create a coarsened graph that can be controlled to be of the   the same size as the graph to be compared. The pipeline of the method is therefore to: first reduce the largest graph to the size of the smallest, then align them using a spectral method, and finally calculate the distance as one of the geodesic distances of the (Riemannian) space of (regularised) Laplacian matrices, in this case the AIRM distance.
The rest of the paper considers this distance, GGD, for the numerical study of GNN properties and whether GGD is a good metric to replace, for example, a GNN for graph classification. The performances reported are adequate. One specific point is that the method can be based on a partial view of the characteristics of the nodes, which brings substantial gains.

**Strengths:**

The article has some good points in its favour:

- the idea is not too complex, the method is feasible and the three steps are well documented in the article, with a sufficient level of explanations and details.

- on the dataset studied, performance is correct and computing times are slightly reduced as compared to state-of-the-art. Also, the scaling is in O(n^3) while previous solutions based on OT have an additional multiplicative log(n) term.

- the topic is useful, in that we have now a lot of articles on GNN, yet it is not always clear how each specific model behaves. The proposed distance appears to behave in a way similar to the output of the GNN called GIN, and gives then some idea about how to compare graphs.

**Weaknesses:**

Despite having some strengths, I find also many weak points in this work, and currently too many to recommend it for acceptation:

* The article does not answer well the following question: is it important to consider a proper Riemannian metric (once the steps of coarsening and alignement) are done ? What would change if one uses anyone of the spectral distances between graphs instead of the third step ?

* 4.3 is supposed to explore a part this question: "CONNECTION BETWEEN GGD AND GRAPH STRUCTURAL MISMATCHES", yet I am not certain to really understand the argument. The theorem provides bound on the generalized eigenvalues used in the distance, that I understand. However, as  it only impacts the extremal eigenvalues, I have uncertainties to whether it really always impact the full AIRM distance. The authors should state more specific elements on that.

* Some choices in the method can be considered as ad hoc, and not fully argument in the text. Examples :

. Why use a Riemannian distance in the third step and never consider this structure for the 2 other ingredients of the method: coarsening and alignement.

. Why use the GRAMPA alignment and not other ones ? (there are several of them with OT, or without)

. How are chosen the parameters ? The choice of $\eta$ for instance does not seem to be in coherence with Lemma 4.1  (1/log n is around 1/3 for graphs with n=20 nodes, like in MUTAG dataset; I don't see why, in A.9, this leads to \eta = 0.5 in the present work).

. Is the proposed coarsening method the best one for the task at hand ? One would have expected something based on the proposed Riemannian approach, no ? Or the use of some already well known coarsening methods (the authors quoted many themselves). Why a new one is needed ?

. The combination of effective resistance and features seems to be completely empirical  (eq. (15)), and should be justified, at least by giving some insights about how it behaves.

* One lacks a summary of the proposed method, in form of a detailed algorithm or a pipeline.

* The numerical examples are too specific: there are not enough datasets tested (4 only) and the graphs in these datasets are always small graphs (average numbers of nodes from 18 to 47) and they correspond to molecules. All that is very specific and somehow it limits in the scope of the work to graphs representing molecules.

* On such small datasets, a separation between train / test / validate data should be expected (here, only train / test is mentioned).

* Many paragraphs of the Appendices are in fact not useful, as they cover well known things. This inflates the article without any valid reason. (See the suggestions underneath)

* Section 3 repeats many things that were already written either in the introduction or that will be presented in greater details in Section 4. I am not certain that it's the best way to present the work.

**Questions:**

* In eq. (14), why are the features put there ? They should only be in (15), shouldn't they ?

* Suggestion: re-write the article to split 3 between the introduction and the Section 4 so as to avoid redundancy.

* Why are std reported for MUTAG in Table 3 and not systematically for other datasets ? Also, check How many significant digits are there, so as to report the numbers correctly in the table.

* Suggestions for the Appendices:

. My feeling is that elements in A.1, A.3 and A.8 do not really add useful elements ; either it's common knowledge (A.1), unrequired additional comments (A.3) which could be in 5.1 (for the references).

. Then, on the Riemannian aspects (Riemannian  being mispelled in title of A.2): A.2 is to short to introduce what it means to a reader not knowing that, hence its not useful for anyone ; A.4 is not needed as everything comes from the fact that AIRM is a distance. Then, the rest follows without any surprise.

. I question also the usefulness of A.7: given that the article does not go far in the direction of Riemannian space, telling that AIRM is the metric of choice seems to be enough (it would be more interesting if the authors would compare various distance using spectral features, and not only two from Riemannian geometry).

. Then, the rest of the appendices are useful.

---

> ### Author Response · Authors · 2024-11-24
>
> Answer to the weaknesses:
>
> 1. Thanks to the reviewer for raising this question. Spectral distances have limitations when calculating the distance between co-spectral graphs (distinct graphs that share the same eigenvalues in their adjacency matrices). Our method, which employs the Riemannian metric, overcomes this limitation by using generalized eigenvalues derived from their modified Laplacian matrices.
>
> 2. Section 4.3 demonstrates the relationship between generalized eigenvalues and graph structural mismatches using the graph cut mismatch. This relationship is not limited to the most extreme generalized eigenvalues but applies to all the generalized eigenvalues computed with both graph Laplacians. Empirical results support our claim. For two graphs of the same size, we observe varying graph cut mismatches for different sets of corresponding nodes. The extreme eigenvalues align with the highest mismatches, followed by the second-largest eigenvalue corresponding to the next largest mismatch. Additionally, there are several node sets where the cuts in both graphs are nearly identical, resulting in a cut mismatch close to 1. This is also reflected in the generalized eigenvalues, where we observe a significant number of unity values. We have included Section A.7 in the supplementary material of our revised draft, demonstrating this example with a graph pair from the MUTAG dataset.
>
> 3.a. The coarsening step is essential for the formation of the Riemannian manifold (a cone of the Symmetric Positive Definite (SPD) matrices of the same sizes), whereas the graph matching is critical for computing the geodesic (shortest) distance between two modified Laplacian matrices. In other words,  without converting two graphs of the ones with the same size, we can not use the Riemannian metric defined for SPD matrices; without doing graph matching, we can not compute the meaningful geodesic distance between two graphs since unmatched graphs will result in much greater distances on the aforementioned Riemannian manifold.
>
> 3.b. Unlike some existing matching methods that use only the extreme eigenvalues and eigenvectors, GRAMPA constructs the similarity matrix using all possible combinations of eigenvalues, thus achieving the most accurate alignment possible. Since the matching procedure in GRAMPA is performed using the eigenvalues and eigenvectors associated with the graph adjacency matrices, the proposed spectrum-preserving graph coarsening step will make sure the node merging step will not significantly impact such eigenvalues and eigenvectors. During our exploration of various graph-matching methods, we found that the spectral method GRAMPA produced better results compared to the others.
>
> 3.c.  We employed an iterative process to determine the optimal value of $\eta$. In Lemma 4.1 of our paper, we referenced the selection method of $\eta$ proposed in the GRAMPA paper [2]. The regularization parameter is recommended to fall approximately within the range $[n^{-0.1}, c/log(n)]$, where n represents the number of nodes, and c is a positive constant that is typically set to unity (1) for approximations discussed later in the GRAMPA paper. For the MUTAG dataset, with an average node count n = 19.79, the range for $\eta$ is [0.335,0.742]. The GRAMPA paper also notes that the accuracy of the matching is not highly sensitive to this parameter within a specific range. For practical datasets, iterative methods are suggested for identifying the optimal value, which we discussed in Appendix A.11 of the revised draft (previously Appendix A.9 in the earlier version).
>
> In our experiments, we tested with several values of $\eta$ and observed the highest accuracy at 0.5. Figure 6 illustrates the impact of $\eta$ on classification accuracy, revealing that the accuracy remains nearly constant for values between 0.2 and 0.5, supporting the GRAMPA paper's claim regarding the parameter's insensitivity.
>
> 3.d. To address the challenge of precisely matching the node counts of two graphs, we developed this spectral coarsening method with the goal of best-preserving graph spectral (structural) properties. Although there are several existing coarsening methods proposed for preserving graph spectral (structural) properties, they cannot control the number of nodes in the reduced graph. Our method is the only approach that allows us to coarsen edges one by one until the desired graph size is achieved, which is crucial for the subsequent graph matching and manifold formation steps.

---

> ### Author Response · Authors · 2024-11-24
>
> 3.e. Thanks to the reviewer for raising this concern. When coarsening an edge, we calculate the effective resistance between the two nodes it connects. If the effective resistance is lower than that of other edges, the edge can be coarsened without significantly compromising the graph's structure. However, when proper node features are available, coarsening two nodes with different features, even if they are structurally close, could disrupt the graph's properties.
> By adjusting the weight, we can control the coarsening process. For instance, if the weight is sufficiently large, the modified effective resistance between nodes with different features will always outweigh that of nodes with similar features, effectively preventing their coarsening. In the highlighted section of 5.1 in the revised paper, we have provided additional clarification.
>
> 4. We agree that providing an algorithmic flow including all the proposed steps would greatly enhance understanding of the processes. In the supplementary section of our revised paper (Appendix A.1), we have included such a flow.
>
> 5. We applied our method to compare with existing classification techniques, using datasets that are commonly benchmarked in graph classification tasks. This benchmarking approach was also inspired by the TMD paper [2].
>
> 6. Assuming the reviewer is referring to the graph classification task using GINs, we employed the method described in the original GIN paper [3]. However, we agree with the reviewer that a train/test/validation split can lead to a more unbiased classification. To address this, we conducted experiments using an 80%-10%-10% split, instead of the 90%-10% split and observed the same results.
>
> 7. We understand the reviewer's concern. For readers with a strong background in Spectral Graph Theory and Graph Neural Networks, these paragraphs may seem unnecessary. However, we included them as supplementary material for those in the general Machine Learning field who may have a limited understanding of Graph Theory.
>
> 8. In our paper, we introduced the concepts of Graph Distances, reviewed related works, and outlined the motivation and contributions of our study in the introduction. Section 3 provided an overview of our methodology and a high-level description of the steps involved, while Section 4 offered a detailed explanation of the methods. In the revised draft, we have modified these three sections to minimize repetition.
>
> Answer to the questions:
>
> 1. The reviewer is correct. In Equation 14, the resistive distance is calculated, while in Equation 15, the modified resistive distance is computed by adding the weighted Euclidean distance between node features. This error has been corrected in the revised version of the paper
>
> 2. In the revised draft, we have modified these three sections to minimize repetition.
>
> 3. We thank the reviewer for highlighting this point. We have revised Table 2 to present the results in a more consistent format.
>
> 4. All the appendix numbers have been updated in the revised paper as we included two additional sections. Here, we will mention the previous appendix numbers for reference.
>
> We understand A.1 (Graph Adjacency and Laplacian Matrices), A.2 (Riemannian Manifold), A.3 (Effective Resistance in Graph Theory), A.8 (Using Normalized Laplacians for GGD Calculation) may be common knowledge for individuals with a background in Spectral Graph Theory and Riemannian Geometry. However, we included these sections as references to assist readers in the broader Machine Learning community who may have limited knowledge of these topics.
>
> AIRM is a Riemannian metric on the Riemannian manifold. Based on this assumption, in A.4 (Detailed Proofs Showing GGD is a Metric), we provided the necessary proofs to establish GGD as a valid distance metric.
>
> In A.7 (Comparison of Two Different Riemannian Metrics for SPD Matrices), we compared the two most well-known Riemannian metrics and explained why AIRM was selected as our metric of choice.
>
>
> [1] Zhou Fan, Cheng Mao, Yihong Wu, and Jiaming Xu. “Spectral graph matching and regularized quadratic relaxations: Algorithm and theory”. In International conference on machine learning, pp. 2985–2995. PMLR, 2020.
>
> [2] Ching-Yao Chuang and Stefanie Jegelka. “Tree mover’s distance: Bridging graph metrics and stability of graph neural networks.” Advances in Neural Information Processing Systems, 35:2944–2957, 2022.
>
> [3] Keyulu Xu, Weihua Hu, Jure Leskovec, and Stefanie Jegelka. “How powerful are graph neural networks?” In International Conference on Learning Representations, 2019.

---

> ### Comment · Reviewer_onsy · 2024-11-25
>
> I have read the various elements of answers, and the revisions made by the authors. I have also read the other reviews and the following discussions. I think that the reviewers have a point about the fact that the authors don't justify enough the novelty of their work w.r.t. the work of Him et al., and their GGD (for instance the need to use a coarsening technique, while Him et al. had another methods).
> As there are also some other points that could benefit from major revisions (deeper than the one currently done), and despite the fact that I acknowledge that the authors tried to revise some of the points and made some efforst toward better clarity, I keep my review as it is.

---

> > ### Author Response · Authors · 2024-11-27
> >
> > In our work, we propose an approach that includes multiple steps. In the final phase, we calculate the distance between SPD matrices in the Riemannian cone using the Riemannian metric, which is parallel to the work of Lim et al. A key contribution is converting graph data into SPD matrices, addressed through graph matching and transformation of Laplacian matrices. Graph matching is crucial, as it ensures node-to-node correspondence, enabling the shortest possible distance on the manifold.
> >
> > Let’s consider an example of two identical graphs with different node indices. We can create this scenario by randomly reassigning the node numbers of the first graph to generate the second graph. As a result, these two graphs will have different adjacency matrices. If we calculate the distance between them at this stage, we will have different SPD matrices and thus will obtain a large positive value. However, after performing the graph matching step, the matrices will align, and the distance will be 0, as expected. This matching step is crucial for determining the shortest distance within the Riemannian cone.
> >
> > To address differences in graph sizes, we use a spectral coarsening method to equalize their sizes. Since this is a spectral approach, it preserves graph properties as much as possible. In contrast, the submatrix adaptation process mentioned by Lim et al. would fail for graph data. Because it means we are throwing away some nodes randomly, which compromises the graph's structural integrity.

---

### Official Review · Reviewer_Z8Nj · 2024-11-04

**Soundness:** 2
**Presentation:** 3
**Contribution:** 2
**Rating:** 3
**Confidence:** 5

**Summary:**

The aim of the authors is to propose a Geodesic Distance (i.e. an spectral-oriented metric) between
graphs. Then they commence by reviewing a couple of classical distances (edit distance and tree-movers)
in order to fill their gaps: lack of globality (this is not well undestood: Edit distance is global but being
aware of this globality is NP-hard) and attribute dependence.

As a result, the classical Laplacian oriented approach emerges. The basic idea is that Laplacian matrices are
not positive definite matrices (PD) but they can be transformed into them by modifying their diagonals. PD is
a requirement to use a Geodesic Distance yet proposed into the literature (Lim et al 2019).

When the graphs do not have the same number of nodes the largest graph is reduced via resistive approaches.

**Strengths:**

-Relate Geodesic Distances to GNN outputs.
-Explain structural mismatches.

**Weaknesses:**

- Lack of originality:

The distance proposed is yet introduced in

"Geometric Distance Between Positive Definite
Matrices of Different Dimensions
Lek-Heng Lim , Rodolphe Sepulchre , Fellow, IEEE, and Ke Ye".
The authors only modify the graph Laplacians to be Positive Definite.

**Questions:**

None.

---

> ### Author Response · Authors · 2024-11-24
>
> Answer to the weaknesses:
>
> The Lim paper [1] does not address the problem of calculating a distance metric between graphs. Our work first performs graph matching and then converts the matched graphs to $n \times n$ real positive symmetric definite (SPD) matrices that naturally form a Riemannian manifold. Then the geodesic distance between data points is calculated using a Riemannian metric. Here graph matching is a crucial step that confirms the infimum distance between data points in the manifold. In the case of graphs of different sizes, we used a resistive distance-based spectral coarsening method to match their sizes and then proceeded with subsequent steps. To our knowledge, no previous work has used this method to determine the distance between graphs. The mentioned paper has influenced our idea generation and literature review, which we have properly cited in our paper. However, our problem statement is fundamentally different.
>
> [1] Lek-Heng Lim, Rodolphe Sepulchre, and Ke Ye. “Geometric distance between positive definite matrices of different dimensions.” IEEE Transactions on Information Theory, 65(9):5401–5405, 2019.

---

> > ### Comment · Reviewer_Z8Nj · 2024-11-25
> > **Comment**
> >
> > The authors need to be aware of the deep relationship between positive define matrices and effective resistances. Actually, we do not need to rely on making Laplacians PD but realizing of the identity between effective resistances an the Laplacian pseudoinverse. See the seminal paper: "Graph Sparsification by Effective Resistances". This paper of "2008" actually poses Laplacian similarity in terms of spectral bounds.

---

> > > ### Author Response · Authors · 2024-11-27
> > >
> > > We are afraid, the concern raised by the reviewer is not relevant to our work. In our approach, effective resistance is used solely for graph coarsening only if the graphs have different node counts. If we have two same-sized input graphs, or once the input graphs are coarsened to have the same number of nodes, we perform graph matching to establish node-to-node correspondence. These matched graphs are then converted into SPD matrices, enabling the use of Riemannian metrics to compute distances in the Riemannian cone. Notably, effective resistance plays no role in the distance calculation phase, and SPD matrices are not required during the coarsening step.

---

### Author Response · Authors · 2024-11-24

We sincerely thank all the reviewers for their valuable questions and suggestions. We have carefully addressed each question and are pleased to upload a revised version of our paper. In this updated version, we have made the following changes:

1. We revised the abstract to emphasize on the development of the distance metric, with stability analysis and graph classification being applications of the metric.
2. We refined the introduction and Section 3 to minimize repetition.
3. We extended section 5.1 to explain the effect of node features on the graph coarsening process.
4. We revised Table 2 to represent the results more consistently.
5. We added an algorithm flow in Appendix A.1 to summarize all the proposed steps.
6. We included Appendix A.7 to demonstrate the relationship between generalized eigenvalues and graph cut mismatches.
7. We corrected Equation 14.

---

### Note · Authors · 2024-12-11

I have read and agree with the venue's withdrawal policy on behalf of myself and my co-authors.